# Subtractive Proteomics and Reverse-Vaccinology Approaches for Novel Drug Target Identification and Chimeric Vaccine Development against *Bartonella henselae* Strain Houston-1

**DOI:** 10.3390/bioengineering11050505

**Published:** 2024-05-17

**Authors:** Sudais Rahman, Chien-Chun Chiou, Shabir Ahmad, Zia Ul Islam, Tetsuya Tanaka, Abdulaziz Alouffi, Chien-Chin Chen, Mashal M. Almutairi, Abid Ali

**Affiliations:** 1Department of Zoology, Abdul Wali Khan University, Mardan 23200, Khyber Pakhtunkhwa, Pakistan; sudaisrahman545@gmail.com; 2Department of Dermatology, Ditmanson Medical Foundation Chia-Yi Christian Hospital, Chiayi 600, Taiwan; 07063@cych.org.tw; 3Institute of Chemistry and Center for Computing in Engineering and Sciences, University of Campinas (UNICAMP), Campinas 13084-862, Brazil; shabirjan427@gmail.com; 4Department of Biotechnology, Abdul Wali Khan University, Mardan 23200, Khyber Pakhtunkhwa, Pakistan; 5Laboratory of Infectious Diseases, Joint Faculty of Veterinary Medicine, Kagoshima University, Kagoshima 890-0065, Japan; 6King Abdulaziz City for Science and Technology, Riyadh 12354, Saudi Arabia; 7Department of Pathology, Ditmanson Medical Foundation Chia-Yi Christian Hospital, Chiayi 600, Taiwan; 8Department of Cosmetic Science, Chia Nan University of Pharmacy and Science, Tainan 717, Taiwan; 9Ph.D. Program in Translational Medicine, Rong Hsing Research Center for Translational Medicine, National Chung Hsing University, Taichung 402, Taiwan; 10Department of Biotechnology and Bioindustry Sciences, College of Bioscience and Biotechnology, National Cheng Kung University, Tainan 701, Taiwan; 11Department of Pharmacology and Toxicology, College of Pharmacy, King Saud University, Riyadh 11451, Saudi Arabia; mmalmutairi@ksu.edu.sa

**Keywords:** *Bartonella henselae*, subtractive proteome analysis, reverse vaccinology, immune simulation, in silico cloning

## Abstract

*Bartonella henselae* is a Gram-negative bacterium causing a variety of clinical symptoms, ranging from cat-scratch disease to severe systemic infections, and it is primarily transmitted by infected fleas. Its status as an emerging zoonotic pathogen and its capacity to persist within host erythrocytes and endothelial cells emphasize its clinical significance. Despite progress in understanding its pathogenesis, limited knowledge exists about the virulence factors and regulatory mechanisms specific to the *B. henselae* strain Houston-1. Exploring these aspects is crucial for targeted therapeutic strategies against this versatile pathogen. Using reverse-vaccinology-based subtractive proteomics, this research aimed to identify the most antigenic proteins for formulating a multi-epitope vaccine against the *B. henselae* strain Houston-1. One crucial virulent and antigenic protein, the PAS domain-containing sensor histidine kinase protein, was identified. Subsequently, the identification of B-cell and T-cell epitopes for the specified protein was carried out and the evaluated epitopes were checked for their antigenicity, allergenicity, solubility, MHC binding capability, and toxicity. The filtered epitopes were merged using linkers and an adjuvant to create a multi-epitope vaccine construct. The structure was then refined, with 92.3% of amino acids falling within the allowed regions. Docking of the human receptor (TLR4) with the vaccine construct was performed and demonstrated a binding energy of −1047.2 Kcal/mol with more interactions. Molecular dynamic simulations confirmed the stability of this docked complex, emphasizing the conformation and interactions between the molecules. Further experimental validation is necessary to evaluate its effectiveness against *B. henselae*.

## 1. Introduction

*Bartonella henselae*, a Gram-negative bacterium, causes cat-scratch disease (CSD) and other clinical manifestations in humans. The attention drawn to this bacterium is due to its ability to establish chronic intravascular infections and its complex interactions with both the host and vector fleas [1]. Among the various strains of *B. henselae*, the Houston-1 strain is of particular importance due to its association with severe disease outcomes and distinct genetic characteristics [2].

Cat-scratch disease is predominantly transmitted to humans via bite or scratch injuries caused by infected cats. The characteristic clinical manifestation of this illness is fever accompanied by self-limiting lymphadenopathy [3]. However, the infection can cause severe clinical manifestations in immune-compromised individuals and in rare instances, in immune-competent patients, including bacillary angiomatosis, hepatic peliosis, and bacillary splenitis. This spectrum of disease underscores the pathogenic potential of *B. henselae* and the importance of understanding its virulence determinants [4]. Inconclusive evidence exists regarding human infection with *B. henselae* via arthropod vectors; however, fragmented studies have identified various species of *Bartonella* in arthropods that utilize humans as atypical hosts [5]. Investigating the proteomic aspects of this strain holds promise for unraveling the underlying mechanisms driving its pathogenicity and host interaction [6]. Moreover, insights were gained based on its complete proteome and its significant medical importance to develop improved therapeutic strategies and potential vaccine candidates to combat *B. henselae* infection [7].

A subtractive proteomics approach is used to identify pathogen-specific proteins that are essential for the pathogen’s survival but are absent in the host [8,9,10]. This approach involves comparing the proteomes of the pathogens and the hosts to identify proteins unique to the pathogens. By subtracting host-specific proteins, subtractive proteomics helps in identifying potential drug targets or vaccine candidates specific to the pathogen [11]. It aids in understanding the pathogen’s biology and virulence factors, while reverse vaccinology is a novel bioinformatics-driven approach that has revolutionized vaccine development by predicting new protein-based vaccine candidates which overcome the current vaccinology limitations. This approach involves analyzing the entire proteome of a pathogen to predict antigenic proteins that could serve as vaccine candidates [12]. By screening the pathogen’s proteome for proteins that are surface-exposed, conserved among strains, and capable of eliciting immune responses, reverse vaccinology identifies potential vaccine targets [13]. This approach has accelerated vaccine development by neglecting traditional methods of isolating and culturing pathogens, allowing for the rapid identification of vaccine candidates for various infectious diseases [14]. Subtractive proteomics and reverse vaccinology have emerged as potent tools for pinpointing drug targets and vaccine candidates. While these methodologies have traditionally been employed independently, they are now being integrated to design innovative drugs and vaccines specifically targeting Gram-negative bacteria [8]. In addition to the proteins, some factors contribute to the attachment of bacteria, such as virulence factors and resistance determinants. Normally, cytoplasmic proteins are targeted for drug development, while proteins found in the membrane or secreted by bacteria are considered for vaccine development [15]. Previously, different membrane-bound proteins have been used to determine antigenic, non-allergenic, and non-toxic epitopes which are capable of being employed in the development of a chimeric subunit vaccine [16]. Therefore, the current study aimed to use subtractive proteomics and a reverse-vaccinology approach to identify novel drug targets and design a multi-epitope vaccine to combat *B. henselae* infections.

## 2. Materials and Methods

### 2.1. Pathogen Proteome Retrieval and Exclusion of Repetitive Sequences

The entire set of proteins of *B. henselae* strain Houston-1 (Assembly GCF_000612965.1) was collected in FASTA from the NCBI (https://www.ncbi.nlm.nih.gov/, accessed on 11 August 2023) and then filtered by CD-HIT (http://weizhongli-lab.org/cd-hit/, accessed on 13 August 2023), using a cut-off value of 0.6 (60%) to eliminate paralogous sequences from the pathogen proteome [17]. Paralogous proteins were excluded and target non-paralogous proteins were used for further study.

### 2.2. Identification of Non-Homologous Proteins

A BLASTp analysis was conducted using an E-value threshold greater than 10^−5^ to pinpoint non-homologous proteins in the pathogen that are distinct from human proteins [18]. Following that, proteins exhibiting similarity to human proteins were eliminated from the dataset.

### 2.3. Identification of Vital Proteins

To pinpoint the vital proteins of the pathogen, a BLASTp search against DEG (http://tubic.tju.edu.cn/deg/, accessed on 28 August 2023) [19] was performed. These vital proteins perform a pivotal role in the pathogen’s survival and growth.

### 2.4. Evaluation of Unique Metabolic Pathways

To evaluate the unique metabolic pathways of *B. henselae* and its host (*Homo sapiens*), the KEGG database was used for comparison (https://www.genome.jp/kegg/, accessed on 05 September 2023). This analysis allowed the identification of pathogen-specific pathways [20]. Then, the KAAS server (https://www.genome.jp/tools/kaas/, accessed on 12 September 2023) [21] was used to provide KO codes indicating the presence of particular proteins in the specific pathways of the pathogen.

### 2.5. Subcellular Localization Analysis

The widely used PSORTb (https://www.psort.org/psortb/, accessed on 16 September 2023) was employed for the exploration of subcellular localization [22] of distinct essential proteins that are present in the unique metabolic pathways of *B. henselae*.

### 2.6. Evaluation of Druggability in Essential and Unique Proteins

Proteins vital for *B. henselae* and non-homologous to humans were analyzed through a BLASTp search against FDA-approved drug targets [23]. Targets that displayed a significant similarity to FDA-approved drug targets were categorized as druggable.

### 2.7. Screening of Gut Microbiota Protein

A BLASTp analysis was conducted against gut flora proteins, employing an E-value cut-off score of 1. Any proteins resembling gut flora proteins were subsequently eliminated [24].

### 2.8. Prediction of Antigenic Membrane Protein

Extracellular membrane proteins were targeted for vaccine development due to their potent epitopes’ ability to enhance immune responses [25]. Proteins with scores >0.4 were considered antigenic, and those with scores <0.4 were labeled non-antigenic, leading to further analysis for potent epitopes.

### 2.9. Protein–Protein Interaction

The String database (string-db.org, accessed on 30 September 2023) was employed, utilizing a higher confidence score of 0.700, to assess the protein–protein interactions associated with the target membrane protein [26].

### 2.10. Prediction of T-Cell MHC-I Epitope

NetCTL was utilized to predict CTL epitopes by combining MHC-binding peptide prediction, proteasome cleavage site prediction, and TAP transport efficiency using an e-value of 0.75 to predict MHC-I epitopes related to the chosen protein [27]. This server selects epitopes based on a high integrated score, considering the intrinsic potential of each peptide [28].

### 2.11. Analysis of Class I Immunogenicity, Antigenicity, Allergenicity, and Toxicity

To evaluate the capability of the epitope to elicit an immune response, the MHC class I immunogenicity prediction tool (http://tools.immuneepitope.org/immunogenicity/, accessed on 10 October 2023) was employed with its default parameter [29]. Epitopes that yielded a positive value in the immunogenicity prediction and were highly antigenic, non-allergic, and non-toxic were subsequently chosen for additional analysis.

### 2.12. Prediction of T-Cell MHC-II Epitopes

For the prediction of T-cell epitope linkages with MHC-II molecules based on IC_50_ values, the SMM method in IEDB (https://www.iedb.org/, accessed on 15 November 2023) was used [30]. A lower IC_50_ value indicates a higher binding affinity of epitopes interacting with MHC-II molecules [31]. The data were uploaded in FASTA format. HLA-DR was selected as the species/locus pair, and alleles were chosen based on the typical length value associated with each species/locus. Other variables remained at their default settings, and the final output format was retrieved as an XHTML table. Allergic and toxic epitopes with values less than 0.4 were excluded.

### 2.13. MHC-Restricted Alleles Clustering

The identification of clusters of MHC-restricted alleles and their corresponding peptides was expedited by using the MHCcluster v2.0 server, offering additional validation to our predictions [32]. This served as a complementary verification to the anticipated MHC-restricted allele evaluation derived from IEDB. The server output encompasses a graphical tree and a static heatmap, providing a visual representation of the active relationships among peptides and alleles [33].

### 2.14. Prediction of B-Cell Epitopes

In addition to MHC-I and MHC-II epitopes for humoral immunity, B-cell epitopes were predicted within the target protein. Humoral immunity is essential for bacterial elimination, was targeted [34]. To find B-cell epitopes, the ABCpred (https://webs.iiitd.edu.in/raghava/abcpred/, accessed on 28 November 2023) identifies linear epitopes with a 75% accuracy rate. Shortlisted epitopes underwent evaluation using various tools, including Kolaskar and Tongaonkar antigenicity, Emini surface accessibility, BepiPred linear epitope prediction, Karplus and Schulz flexibility, and Chou and Fasman β-turn prediction [35,36,37,38]. The protrusion index (PI) scoring system, employed in this analysis, calculates the average value over epitope residues based on an ellipsoidal approximation of the protein’s 3D shape. Higher scores highlight increased solvent accessibility, indicating potentially crucial regions for epitope recognition.

### 2.15. Design of the Multi-Epitope Vaccine Construct

The filtered epitopes were merged to construct a multi-epitope vaccine construct. Initially, an adjuvant (β-defensin) was connected with the epitope by an EAAAK linker, and MHC-I, MHC-II, and B-cell epitopes were linked by Gly-Pro-Gly-Pro-Gly (GPGPG) linkers [39]. β-defensin served as an adjuvant to boost the immune response and linkers were added to maintain the structural integrity and prevent self-binding [40].

### 2.16. Antigenicity, Allergenicity, and Solubility Evaluation of the Designed Vaccine Construct

For antigenicity analysis, VaxiJen (https://www.ddg-pharmfac.net/vaxijen/VaxiJen/VaxiJen.html, accessed on 5 December 2023) was employed [41]. AllerTOP v2.0 (https://www.ddg-pharmfac.net/AllerTOP/, accessed on 5 December 2023) predicted allergenicity [42], and SOLpro (https://scratch.proteomics.ics.uci.edu, accessed on 5 December 2023) was used to assess and evaluate the water solubility of the constructed vaccine [43]. The outcome was anticipated, achieving an accuracy of 74% with the corresponding probability of ≥0.5.

### 2.17. Secondary and Tertiary Structure Predictions, Refinement, and Validation of the Designed Vaccine Construct

PDBsum (https://www.ebi.ac.uk/thornton-srv/databases/pdbsum/Generate.html, accessed on 12 December 2023) [44] was employed to predict the secondary structure of the multi-epitope vaccine construct. The tertiary structure was predicted by the 3Dpro server (https://scratch.proteomics.ics.uci.edu/casp6_results.html, accessed on 23 December 2023) which incorporates predicted structural features and statistical terms based on data from the Protein Data Bank (PDB) into its energy function [45]. Subsequently, the initial model was refined through the GalaxyRefine tool (http://galaxy.seoklab.org/cgi-bin/submit.cgi?type=REFINE, accessed on 28 December 2023) and further validated by using the empirical R-factor for the validation of protein structures (ERRAT), Ramachandran plot, and ProsaWEB.

### 2.18. Physiochemical Properties of the Designed Vaccine Construct

The physicochemical properties of the designed multi-epitope vaccine construct were assessed using ProtParm (https://web.expasy.org/protparam/, accessed on 5 January 2024) [46].

### 2.19. Disulfide Engineering of the Designed Vaccine Constructs

To aid in protein folding and enhance structural stability, Disulfide by Design (DbD) v2.13 was utilized (http://cptweb.cpt.wayne.edu/DbD2/, accessed on 9 January 2024) for disulfide engineering of the designed vaccine construct [47]. This approach facilitates the creation of a stable modeled structure for the vaccine construct. Residue pairs were selected based on specific criteria: chi3 value of −87° or +97° ± 30 and energy value <2.2 kcal/mol [48].

### 2.20. Molecular Docking of the Designed Vaccine Construct with Human Toll-Like Receptor 4 (TLR4)

Molecular docking was employed using the ClusPro server (https://cluspro.org/, accessed on 14 January 2024) to explore binding interactions within the vaccine construct and human toll-like receptor 4 (TLR4) (PDB: 3FXI) [49]. ClusPro stands out as a comprehensive protein–protein docking web-server, utilizing a hybrid docking method. This method draws a hybrid docking algorithm from the experimental data of the substrate-binding sites of protein and small-angle X-ray scattering. Through this approach, ClusPro generates ten docking models. Molecular interactions within the vaccine–TLR4 complex were illustrated using PDBsum (https://www.ebi.ac.uk/thornton-srv/databases/pdbsum/, accessed on 20 January 2024).

### 2.21. Molecular Dynamics Simulation

Three systems were constructed, including the vaccine construct, toll-like receptor 4 (TLR4), and vaccine–TLR complex, and simulations were performed under constant conditions of 300 K temperature and 1 bar pressure, employing the V-rescale thermostat and the Parrinello–Rahman barostat. The systems were enclosed within cubic boxes and immersed in a solvent of TIP3P water molecules, utilizing periodic boundary conditions. To counterbalance the inherent positive charge of proteins, Na^+^ ions were introduced. Covalent bond constraints were applied using the LINCS method with 2 fs integration steps, and electrostatic interactions were computed via the particle-mesh Ewald method. The energy minimization of proteins solvated in water and Na^+^ was performed with the steepest descent algorithm, requiring approximately 50,000 steps that were adjusted according to the specific needs of each system. Subsequently, the minimized protein structures underwent a 1 ns canonical ensemble constant number of particles, volume, and temperature (NVT) equilibration step, followed by a 5 ns canonical ensemble constant number of particles, pressure, and temperature (NPT) equilibration step. The equilibrated system then proceeded to the production phase of 100 ns molecular dynamics (MD). All the molecular dynamics simulations described in this article were performed with GROMACS version 2023.2 [50]. The trajectory data of the system were analyzed using visual molecular dynamics (VMD) [51] and ChimeraX (version 1.6.1) [52]. The root mean square deviation (RMSD) and root mean square fluctuation (RMSF) plots were generated using Grace (version 5.1.25) [53].

### 2.22. Discontinuous B-Cell Epitope Prediction

The ElliPro tool of the IEDB resource was utilized for precise prediction of conformational epitopes, notably discontinuous B-cell epitopes, accessible at http://tools.iedb.org/ellipro/, accessed on 29 January 2024 [54].

### 2.23. Simulation of Immunity

The simulations were executed using the C-ImmSim tool (https://kraken.iac.rm.cnr.it/CIMMSIM/, accessed on 5 February 2024) with its default parameters, to evaluate the immunogenic value of the designed multi-epitope vaccine construct. This tool incorporates real-life immune responses and interactions, as well as machine learning, through the use of a position-specific scoring matrix (PSSM) [55]. The output of this tool is provided based on the immunostimulatory activities in anatomical regions, comprising bone marrow, thymus, and lymph nodes. The time steps in the C-ImmSim web tool (with default parameters) were set at 1, 42, and 84, with each time step equivalent to 8 h, and the first step representing the injection at time = 0. The time interval between two injections (a total of three injections) was 4 weeks [56].

### 2.24. Codon Optimization of the Designed Multi-Epitope Vaccine Construct and Its Virtual Cloning

The Java Codon Adaptation Tool (JCat) (https://www.jcat.de/, accessed on 10 February 2024) was pivotal in aligning the codon function of the designed multi-epitope vaccine with that of the *E. coli* host strain [57]. The constructed vaccine underwent reverse translation to DNA, and adjustments were made to synchronize the codon usage with the preferences of *E. coli*. This adaptation process relied on codon adaptation index values, calculated through a specific algorithm. Following adaptation, the gene sequence of the ultimate designed multi-epitope vaccine construct was inserted into the *E. coli* pET-28a (+) vector using Snapgene, accessed on 12 February 2024 [58], ensuring the optimized depiction of the designed vaccine construct.

### 2.25. Prediction of the mRNA Structure Encoding the Multi-Epitope Vaccine Construct

The predicted secondary structure of the mRNA of the designed vaccine construct and minimum free energy (MFE) was predicted through the utilization of the RNAfold server (http://rna.tbi.univie.ac.at//cgi-bin/RNAWebSuite/RNAfold.cgi, accessed on 20 February 2024) [59]. The principal parameter of interest was the minimum free energy, expressed in Kcal/mol, with lower values indicating greater stability in the mRNA folding structure [60].

## 3. Results and Discussion

### 3.1. Pathogen Proteome RETRIEVAL, filtration, and Non-Host Homolog Protein Identification

A subtractive proteomics approach that involves analyzing the entire proteome using various online databases and computational tools was utilized to predict drug targets and vaccine candidates against *B. henselae* [61]. The entire proteome of *B. henselae* was retrieved followed by the CD-HIT server [8] to exclude paralogous sequences. A collection of 1213 non-paralogous proteins were curated which are vital for pathogen survival [62]. These proteins were subjected to a BLASTp search against the host (*H. sapiens*), identifying 827 non-homologous proteins essential for further analysis (Table 1).

### 3.2. Identification of Essential Proteins, Unique Metabolic Pathways, and Subcellular Localization

Essential proteins are crucial for sustaining fundamental cellular processes and pathogen viability in microorganisms. To find these essential proteins, a BLASTp search of non-homolog proteins against the Database of Essential Genes (DEG) [63] was employed, resulting in the determination of 153 proteins in *B. henselae*’s metabolic pathways. To identify unique metabolic pathways, a comparison was performed between the metabolic pathways of *B. henselae* and the host organism (*H. sapiens*) [64]. This comparison unveiled 24 metabolic pathways exclusive to *B. henselae* shown in Appendix A, with 20 unique proteins, demonstrating no similarity to the host (*H. sapiens*) metabolic pathways shown in Appendix A. These unique metabolic pathways encompassed lysine biosynthesis, peptidoglycan biosynthesis, lipopolysaccharide biosynthesis, two-component systems, and phosphotransferase systems (PTSs) (Figure 1A), presenting potential targets for drug and vaccine development, offering promising avenues for combating this pathogen [65]. By analyzing these proteins, vital processes of *B. henselae* can be potentially disrupted, leading to the development of effective interventions against this pathogen.

Determining the intracellular localization of proteins is pivotal for identifying potential drug targets and vaccine candidates. The PSORTb server was utilized [66] to identify cytoplasmic proteins that are used as prospective drug targets for pharmacological applications and outer membrane proteins as promising candidates for vaccine development. Out of the 20 target proteins, the subcellular localization predictions revealed that 12/20 (60%) are cytoplasmic, 7/20 (35%) are present in a cytoplasmic membrane, and 1/20 (5%) were unknown (WP_011180128.1 KpsF/GutQ family sugar-phosphate isomerase) in *B. henselae* (Figure 1B).

### 3.3. Assessing Druggability, Virulency, and Screening of Gut Microbiota Proteins

To assess the druggability analysis, a BLASTp analysis of unique essential proteins against FDA-approved drugs was performed [67]. Nine proteins from *B. henselae* were predicted to be druggable, which are undecaprenyldiphospho-muramoylpentapeptide beta-N-acetylglucosaminyltransferase, UDP-N-acetylmuramoyl-L-alanyl-D-glutamate--2,6-diaminopimelate ligase, 3-deoxy-8-phosphooctulonate synthase 3-deoxy-manno-octulosonate cytidylyltransferase, UDP-N-acetylmuramate-L-alanine ligase, penicillin-binding protein 1A, sigma-54 dependent transcriptional regulator, PAS domain-containing sensor histidine kinase (ATP-binding protein), and phosphoenolpyruvate--protein phosphotransferase. The screening of virulent proteins has emerged as a prominent approach for predicting therapeutic targets. To check virulency, a BLASTp analysis of the nine predicted target proteins was performed against the VFDB server [68]. Among the nine target proteins, four proteins were virulent except undecaprenyldiphospho-muramoylpentapeptide beta-N-acetylglucosaminyltransferase, UDP-N-acetylmuramoyl-L-alanyl-D-glutamate-2,6-diaminopimelate ligase, 3-deoxy-8-phosphooctulonate synthase, UDP-N-acetylmuramate-L-alanine ligase, penicillin-binding protein 1A, and phosphoenolpyruvate--protein phosphotransferase. Target proteins were BLASTp screened against gut metagenomic proteins to exclude those from the human gut flora. Out of the nine target proteins, six exhibited no similarity with the human host’s gut metagenome, establishing them as the final target proteins (Table 2). The six target proteins included 3-deoxy-8-phosphooctulonate synthase, UDP-N-acetylmuramate-L-alanine ligase, 3-deoxy-manno-octulosonate cytidylyltransferase, sigma-54 dependent transcriptional regulator, PAS domain-containing sensor histidine kinase (ATP-binding protein), and phosphoenolpyruvate--protein phosphotransferase. These vital, non-homologous, and virulent target proteins represent promising candidates for drug targeting and vaccine development.

### 3.4. Prediction of Antigenic Membrane Protein and Its Interactions with other Proteins

Among the six identified target proteins from gut flora [69], the PAS domain-containing sensor histidine kinase protein was identified as an outer membrane-bound protein with antigenic properties (Antigenicity score: 0.4060), which is found in the two-component system pathway, a signaling mechanism commonly found in bacteria. It involves two key components—a sensor kinase and a response regulator. The sensor kinase detects environmental signals and phosphorylates itself, then transfers the phosphate group to the response regulator, triggering a cellular response. The PAS domain-containing sensor histidine kinase protein shows strong interactions with ompR, a regulator responsive to osmolarity changes, modulating outer membrane protein expression [70], feuP (two-component system regulatory protein), and other proteins like divk_1 and 2, ctrA, and various histidine kinases (CDO46385.1, CDO46063.1, CD47015.1, CDO47413.1, and CDO46446.1). These interactions are crucial for bacterial signal transduction and cell cycle regulation [71], potentially influencing cyclic-di-GMP signaling [72] Appendix A. These proteins play a pivotal role in ensuring the survival of *B. henselae* and their significance is integral to the bacterium’s viability. Hence, modulating PAS domain-containing sensor histidine kinase proteins not only affects their functionality but also impacts the associated proteins.

### 3.5. Prediction of MHC-I Epitopes, Class I Immunogenicity, Antigenicity, and Non-Toxicity Analysis for Designing the Multi-Epitope Vaccine Construct

The NetCTL revealed 14 epitopes within the PAS domain-containing sensor histidine kinase (ATP-binding protein) which are shown in Appendix A [73]. Out of 14, 7 epitopes were highly antigenic, non-allergic, and non-toxic. The affinity of binding among the MHC complex and TCR4 was analyzed to evaluate their immunogenic potential, with a high antigenicity score indicating a potent capability to activate inexperienced T cells and provoke a cellular immune response [74]. Subsequently, the class I immunogenicity showed that out of the 7 epitopes, 4 exhibited positive values, and these were targeted for additional analysis (Table 3).

### 3.6. Prediction of MHC-II Epitopes for Designing the Multi-Epitope Vaccine Construct

Beyond MHC-I epitope prediction, the selected proteins underwent MHC-II binding prediction [75]. Out of the 55 MHC-II epitopes which are presented in Appendix A, a subset of 5 epitopes was selected. Epitopes were subsequently screened for toxicity, allergenicity, and antigenicity to finalize their selection for further analysis (Table 4).

### 3.7. Assessment of MHC Restriction and Cluster Analysis

A comprehensive analysis was carried out on MHC-I and MHC-II restricted alleles, considering their IC_50_ values. All predicted epitopes underwent individual scrutiny for MHC interaction analysis [76], encompassing four MHC-I epitopes and five MHC-II epitopes. Subsequently, the alleles participating in these interactions underwent meticulous reassessment through cluster analysis, resulting in the generation of a heatmap illustrating MHC-I and MHC-II as shown in Appendix A interactions, along with a dynamic tree [77]. Red zones on the heat map signify stronger interactions, while yellow zones indicate weaker interactions among clusters of MHC molecules.

### 3.8. Identification of B-Cell Epitopes for Designing the Multi-Epitope Vaccine Construct

The ABCpred identified 19 linear B-cell epitopes of 20 lengths each, with a precision of 75% as shown in Appendix A. From these, 7 antigenic, non-allergenic, and non-toxic epitopes were selected for the construction of the multi-epitope vaccine (Table 5). The chosen epitopes were determined using Chou and Fasman β-turn prediction, Emini surface accessibility, Karplus and Schulz flexibility, Kolaskar and Tongaonkar antigenicity, and linear B-cell epitope prediction using BepiPred [78]. The predicted B-cell epitopes for the PAS domain-containing sensor histidine kinase protein are in Appendix A.

### 3.9. Formulation of the Epitope-Based Subunit Vaccine

Adjuvant is an essential component of vaccines that enhances the immune response [79]. The inclusion of β-defensin as an adjuvant in the vaccine aims to leverage its immunomodulatory properties [80], enhancing antigen presentation and immune responses, thereby improving vaccine efficacy. An EAAAK linker was used to connect the adjuvant with the vaccine construct. The chosen four MHC-I epitopes, five MHC-II epitopes, and seven B-cell epitopes were connected by GPGPG linkers [81]. The resulting vaccine construct is 367 amino acid residues in length.

### 3.10. Allergenicity, Solubility, Antigenicity, and Physiochemical Features of the Designed Multi-Epitope Vaccine Construct

The construct was generated and assessed for antigenicity, solubility, and allergenicity [82]. Our findings reveal a high antigenicity with a score of 0.9948 [83] and predict the construct as non-allergenic [84]. Physicochemical properties were analyzed [85], revealing a hypothetical isoelectric point (pI) of 9.34, a Gravy value of −0.377, and a molecular weight of 38.84 kDa [86]. The system demonstrated structural stability with an instability index of 32.68 and a thermostable aliphatic index of 75.59. The vaccine has a half-life exceeding 20 h in yeast cells and over 10 h in *E. coli* (in vivo). SOLpro predicts a high solubility (0.9195), indicating favorable heterologous expression in *E. coli* (Table 6).

### 3.11. Secondary Structure Prediction of the Designed Multi-Epitope Vaccine Construct

The PDBsum tool was used to determine the secondary structure of the designed vaccine construct, the distribution of α-helices, extended strands, β-turns, and random coils within the predicted secondary structure are shown in Appendix A and (Figure 2) [87].

### 3.12. In Silico Tertiary Structure Prediction, Its Refinement, and Validation.

The 3D structure of the designed vaccine construct was generated using 3Dpro (Figure 3A). In 3DPro, energy functions incorporate statistical terms based on predicted structural features and Protein Data Bank knowledge. During protein modeling, a set of fragment replacements and random perturbations is used to model the target protein [88]. The modeling strategy employs simulated annealing with linear cooling and decisions are made accordingly based on these strategies. As a result, multiple models were generated using random seeds, and their energies were calculated [89]. Finally, the model with the lowest energy was selected and then underwent refinement. The refined construct demonstrated a GDT-HA 0.9775, RMSD 0.334, MolProbity 2.013, Clash score of 12.6, Poor rotamers 0.4 and Rama favored 94.0, and a ProSA-web Z score of −3.98, surpassing the mean Z score observed in similar natural protein (Figure 3C). ProSA-web affirmed the construct’s accuracy by analyzing energy as a role of amino acids in the protein structure (Figure 3D). For comprehensive validation, PROCHECK conducted a Ramachandran analysis, confirming that 92.3% of residues occupied the most favorable (red) region, 7.4% were in the additional allowances (yellow) region, and 0.4% were in the generous allowances (pale yellow) area (Figure 3B). ERRAT assessed the total quality of the vaccine’s 3D structure, assigning it a score of 81.4% (Figure 3E). These assessments collectively contribute to the validation and enhancement of the structural vaccine construct.

### 3.13. Disulfide Engineering for Structural Stability of Vaccine Constructs

To stabilize the structural integrity of the modeled vaccine construct, inter and intra-chain disulfide bonds were evaluated by DbD2. A total of 32 residue pairs were identified as suitable candidates for disulfide modifications [90]. Only two residue pairs were selected based on specific criteria, including an energy score of less than 2.2 kcal/mol and a χ^3^ angle between −87° to +97°. Consequently, four mutations were introduced in the residue pairs. For the TYR174-LYS194 residue pair, the energy score was 1.61 kcal/mol with a χ^3^ angle of +89.87°. Conversely, for the GLY276-ALA290 residue pair, the energy score was 1.76 kcal/mol, and the χ^3^ angle was −89.39° (Appendix A).

### 3.14. Molecular Docking and Interaction of the Multi-Epitope Vaccine Construct with the TLR4 Receptor

Examining the antigenic potential and triggering immune response requires an exploration of the molecular interaction between the constructed vaccine and its targeted human immune receptor [91]. The TLR4 immune receptor is essential for recognizing pathogenic proteins and inducing inflammatory cytokines against various infections. In this study, molecular docking was performed using ClusPro [49]. ClusPro is a widely used protein–protein docking server that predicts the binding modes of protein complexes based on the structures of individual proteins and the interaction between the vaccine construct and the TLR4 receptor [92]. The program generates 30 various clusters and ranks them by energy level. There were −1047.2, −1040.4, −1024.4, −978.6, −971.9, −965.0, −937.8, −929.8, and −903.2 kcal/mol of energy in the ten top clusters. The best group with the lowest energy of −1047 kcal/mol was selected (Figure 4) showing a high affinity for attachment. In-depth analysis revealed specific interactions, including 3 salt bridges, 21 hydrogen bonds, and 225 nonbonding interactions with the constructed TLR4 complex [93]. These results underscore a potential vaccine construct that can specifically link with the immune receptors and potentially trigger an immune response against cat-scratch disease.

### 3.15. Molecular Dynamic Simulation

Continuous advancements in simulation algorithms have positioned molecular dynamics (MD) simulations as indispensable tools in the development of innovative therapeutic approaches [94]. In this investigation, MD simulations were applied to the three systems consisting of the vaccine construct, TLR4 receptor, and vaccine–TLR structure complex in an explicit water environment, extending each for 100 ns. To ascertain the equilibration at 300 K, the deviation of backbone atoms was assessed through the root mean square deviation (RMSD) (Figure 5A). The outcomes showed that the complex simulation adequately achieved equilibration after 22 ns and remained in equilibrium until 100 ns, showing consistent stability in backbone behavior. To gain insights into the significance of specific residues within the three systems, the root mean square fluctuation (RMSF) was examined. The analysis, presented in (Figure 5B), revealed substantial fluctuations in Residues 240–290 of the vaccine construct. This observation implies that the conformational changes in these specific residues may play a pivotal role in binding to the TLR receptor. These findings elucidate the dynamic nature of the vaccine–TLR complex.

### 3.16. Prediction of Discontinuous B-Cell Epitopes

Antibodies target conformational epitopes, notably discontinuous B-cell epitopes, which are crucial for pathogen neutralization. Ellipro identified four discontinuous B-cell epitopes, encompassing a total of 184 residues with scores varying from 0.61 to 0.80 [95] as shown in Figure 6 and Appendix A.

### 3.17. Simulation of Immunity

C-ImmSim revealed a favorable Th2-biased immune response following a single injection without the addition of lipopolysaccharide (LPS) for 1000 antigen molecules [96]. This simulation aimed to determine whether a single dosage could sustain responses for the first month following vaccination [97]. These settings helped to verify that the designed vaccine candidate could induce immune responses persistently in the absence of a booster dose [98] (Figure 7). In Figure 7A, C-ImmSim results revealed an extraordinary development of antibody levels (IgG1, IgM, IgG2) during secondary and tertiary reactions, correlating with diminishing antigen concentrations. Figure 7B illustrates the humoral immune reaction presented by raised levels of IgM and memory B cells enduring B-cell isotypes, confirming the development of memory B cells and their switching capability [99]. The memory development in T-helper and cytotoxic T-cell populations, essential for complementing the immune response, is evident in Figure 7C. There is a clear increase in macrophage activity, coupled with a valuable proliferation of dendritic cells after the immunological response which is shown in Figure 7D. Figure 7E shows enhanced levels of IFN-γ and IL-2. The rise in cytokine scales resulted in higher risks according to the Simpson Index D, posing challenges during the immune response [100]. Notably, these predictions were found compatible with the induced quantity of IFN-γ formed upon immunotherapy by the designed multi-epitope chimera as predicted by C-ImmSim.

### 3.18. Codon Optimization and Virtual Cloning

The expressive capability of the designed vaccine construct was assessed [101]. According to the JCat results for the refined cDNA, the vaccine construct showed CIA values of 1.0 and a GC content ranging from 54.42%, falling within the ideal range for favorable expression in the E. coli K12 vector [102]. The elevated gene sequence of the vaccine construct was anticipated to efficiently integrate into the widely employed pET-28a (+) plasmid, with an overall length of 3651 bp (Figure 8).

### 3.19. Prediction of mRNA Structure Durability in the Designed Vaccine Construct

The RNAfold and JCat analyses revealed a shared centroid secondary structure conformation with minimum free energy (MFE) for the designed vaccines’ mRNA structures [103] (Figure 9). Significantly, the opposing energy values, particularly for the *B. henselae* vaccine (−346.40 kcal/mol), affirm the stability and resilience of the vaccines’ in vivo mRNA forms.

## 4. Conclusions

This study employed subtractive proteomics and reverse-vaccinology approaches to identify key target proteins for the identification of novel drug targets and design of a multi-epitope vaccine against *Bartonella henselae*. Initially, the proteome of the *B. henselae* strain Houston-1 was retrieved and assisted in shortlisting based on their non-redundancy, non-host homology, essentiality, and druggability. Five druggable target proteins were identified as novel, including 3-deoxy-8-phosphooctulonate synthase, UDP-N-acetylmuramate-L-alanine ligase, 3-deoxy-mannooctulosonate cytidylyltransferase, sigma-54 dependent transcriptional regulator, and phosphoenolpyruvate--protein phosphotransferase, and one membrane protein (the PAS domain-containing sensor histidine kinase protein was identified for multi-epitope vaccine construction). B-cell and T-cell epitopes were predicted from the target membrane protein to induce humoral and cell-mediated immunity. Adjuvants and linkers were strategically incorporated to enhance stability, effectiveness, and immunogenicity. The proposed vaccine demonstrated favorable structural, physicochemical, and immunological attributes, including stable binding with TLR-4 receptors. In silico immune simulations indicated a promising in vivo immunogenicity. Reverse translation and codon optimization facilitated effective declaration and stability in *E. coli.* Further experimental validation is vital to ensure the assurance and efficacy of the designed vaccine in animal models.

## Figures and Tables

**Figure 1 bioengineering-11-00505-f001:**
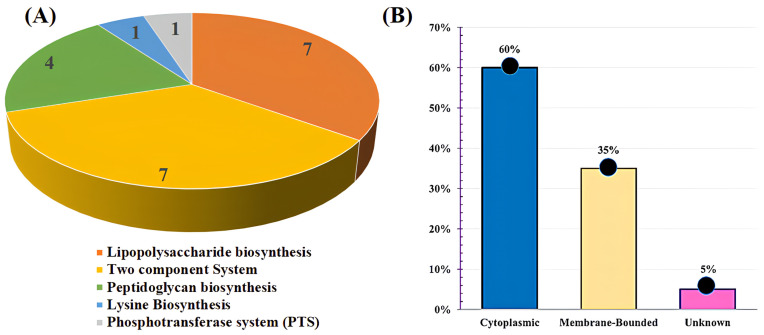
(**A**) Graphical representation showing the number of unique metabolic proteins distributed across different metabolic pathways. (**B**) The percentage of unique metabolic proteins identified in *B. henselae* specific metabolic pathways was determined using PSORTb.

**Figure 2 bioengineering-11-00505-f002:**
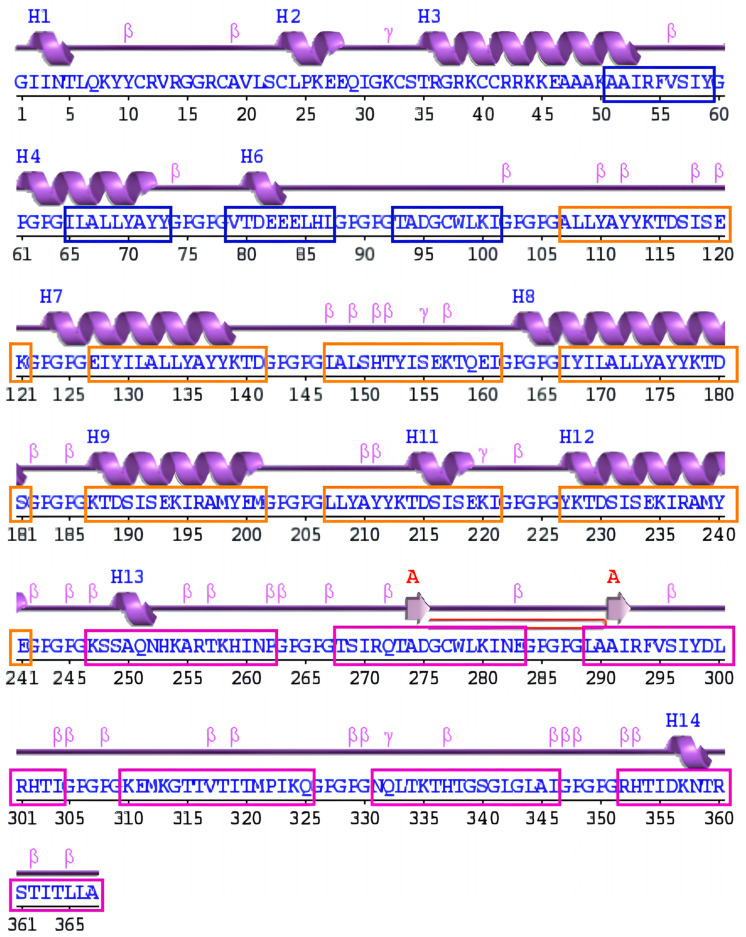
Schematic diagram of the protein’s secondary structure by the PDBsum tool; strands (pink arrows), helices (purple springs), and other motifs in red (e.g., β-hairpins, and γ-turns), and the distribution of MHC-I (blue box), MHC-II (orange box), and B-cell (purple box) epitopes within the predicted secondary structure of the vaccine construct.

**Figure 3 bioengineering-11-00505-f003:**
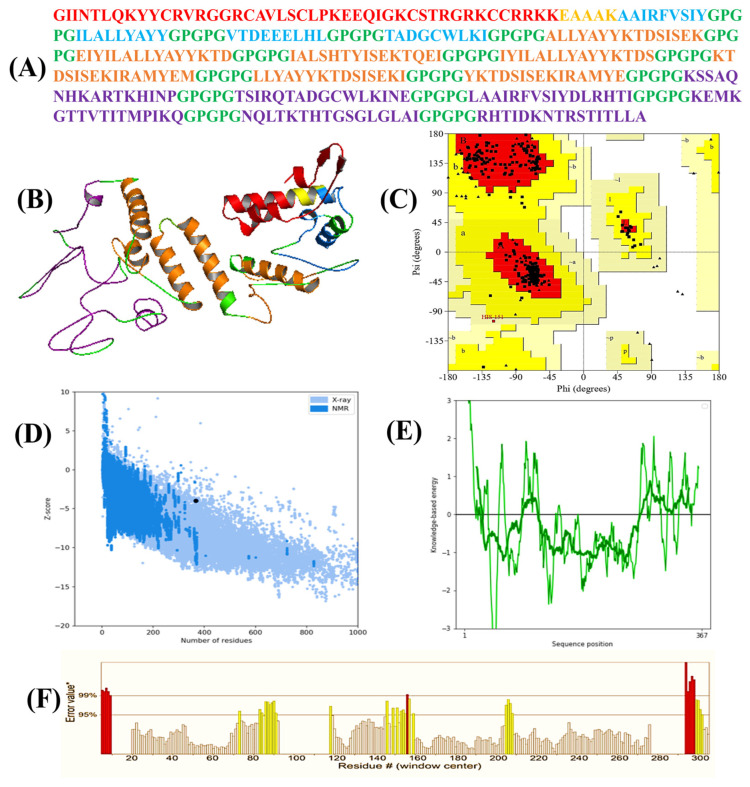
(**A**) Amino acid sequences of the vaccine construct. The adjuvant is colored red, MHC−I epitopes are colored blue, and MHC−II epitopes are colored brown, B-cell epitopes are colored purple, and linkers are colored green. (**B**) Three−dimensional (3D) structure of the vaccine construct. (**C**) In the Ramachandran plot of the refined 3D model generated by the PROCHECK, the red-colored regions are the most favored, the dark yellow and light yellow regions are the additional allowed and generously allowed regions, and the white regions are the disallowed regions. (**D**,**E**) The Z-score plot of the refined 3D model generated by ProSA-web. (**F**) The region in the structure are rejected at the 99% level are shown in red color, also the regions of the structure rejected at 95% confidence level are shown in yellow color which is generated by ERRAT.

**Figure 4 bioengineering-11-00505-f004:**
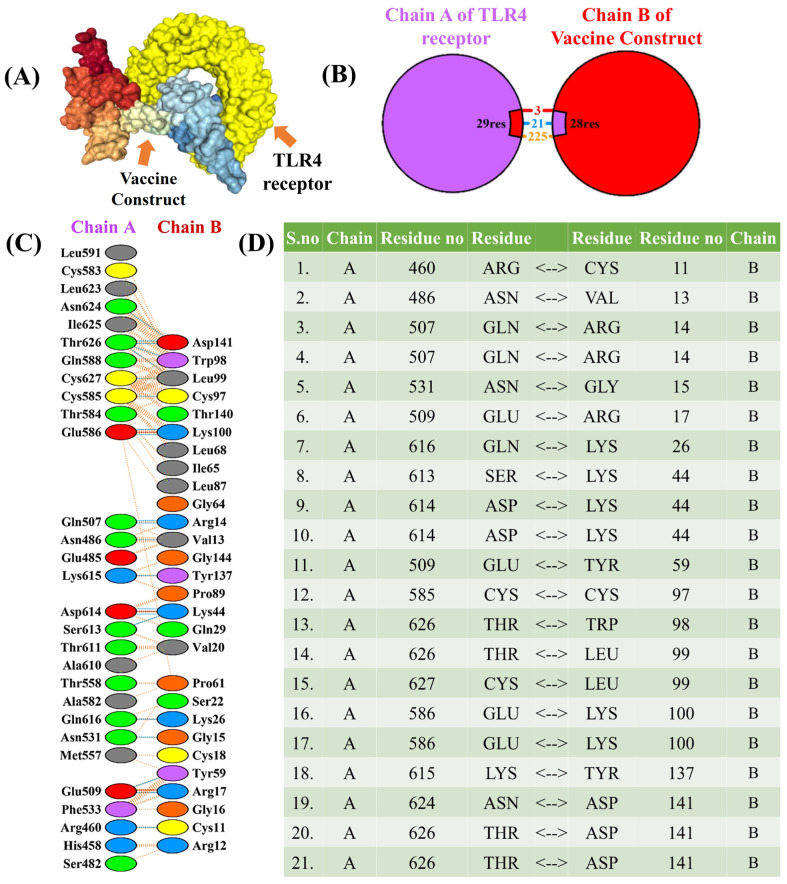
(**A**) 3D visualization of the docked complex between the vaccine construct and TLR4 receptor. (**B**,**C**) The interactions of chain-A of the TLR4 receptor and chain-B of the constructed vaccine. (**D**) Hydrogen-bond interactions among chain-A of TLR4 and chain-B of the constructed vaccine.

**Figure 5 bioengineering-11-00505-f005:**
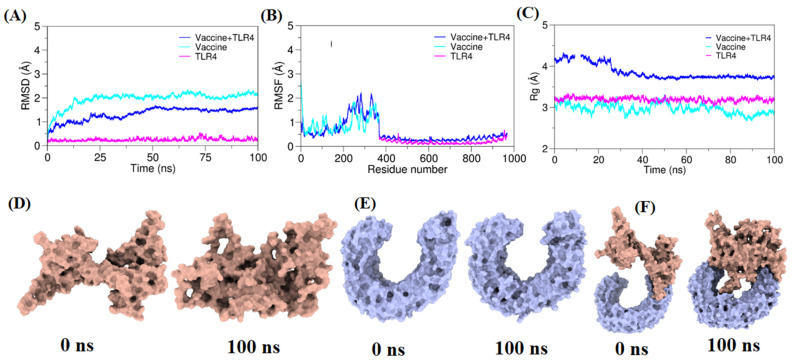
The molecular dynamic plots for TLR4 and vaccine construct complex. (**A**) Root mean square deviation (RMSD) plot of the vaccine–TLR4 complex. (**B**) Root mean square fluctuation (RMSF) plot of the vaccine–TLR4 complex. (**C**) Radius of gyration (Rg) plot for the vaccine–TLR4 complex. The left panel from 0 ns indicates the initial structure of the moieties, and the rightmost 100 ns panel indicates the final conformation of the moieties. (**D**) Represents 0 ns and 100 ns frames from vaccine construct simulation; (**E**) Represents 0 ns and 100 ns frames from TLR4 simulation; (**F**) Represents 0 ns and 100 ns frames from vaccine–TLR4 simulation.

**Figure 6 bioengineering-11-00505-f006:**
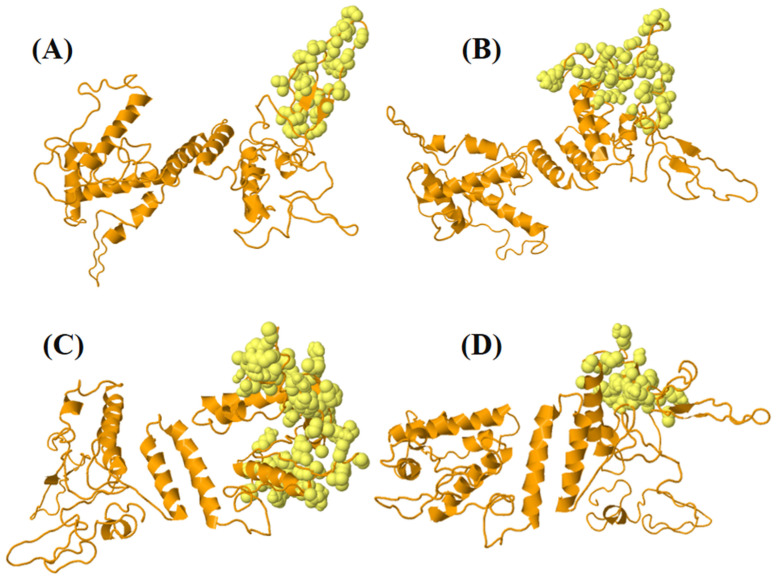
The figure depicts four Ellipro-predicted discontinuous B-cell epitopes present in the vaccine construct: (**A**) 35 residues (AA259-AA294) with score 0.804; (**B**) 45 residues (AA316-AA361) with score 0.728; (**C**) 80 residues (AA1-AA142) with score 0.694; (**D**) 24 residues (AA211-296) with score 0.617; the highlighted orange-colored cartoon illustrates the vaccine construct and the light-yellow spheres show discontinuous B-cell epitopes.

**Figure 7 bioengineering-11-00505-f007:**
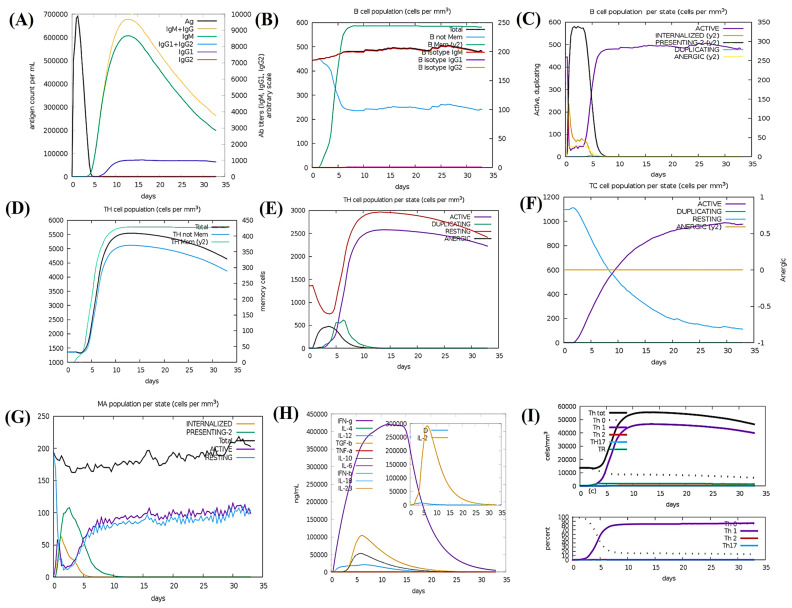
C-ImmSim immunization simulation for the multi−epitope constructed vaccine: (**A**) Immunoglobulin generation, shown by color peaks. (**B**) Cell population indicating increased B-cell types and class−switching potential. (**C**) Population results per state of B cells. (**D**) Evolution of T−helper cells, and (**E**) Population per state of T−helper cells. (**F**) Generation of cytotoxic T cells. (**G**) Population of macrophages per state. (**H**) Induction of cytokines and interleukins, with increased IFN−γ and IL−2 formation after vaccination. (**I**) Th1−implemented immune response.

**Figure 8 bioengineering-11-00505-f008:**
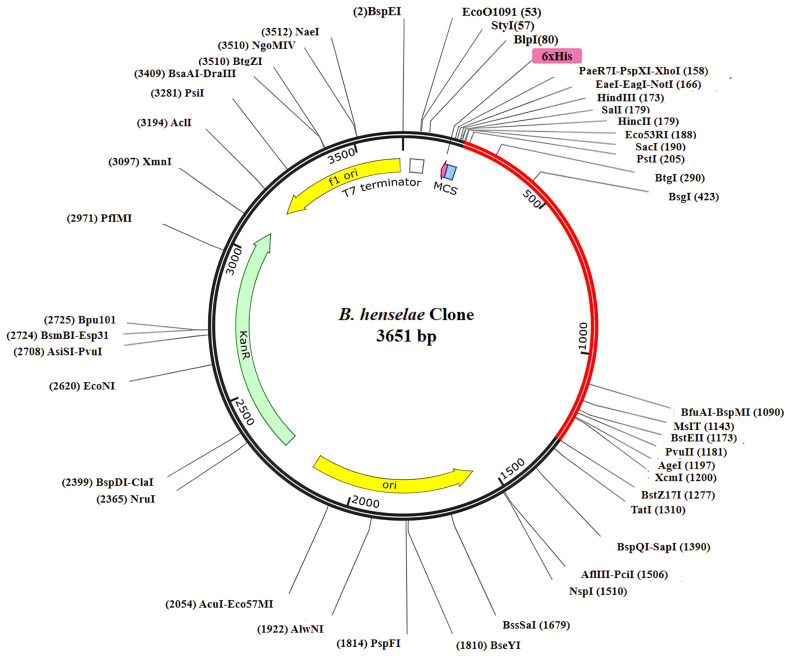
The reverse translated primary DNA sequence of the final multi-epitope vaccine construct was conjoined into the E. coli vector (pET-28a +) through in silico cloning. The vaccine construct is represented in a red color, while the black color indicates the plasmid.

**Figure 9 bioengineering-11-00505-f009:**
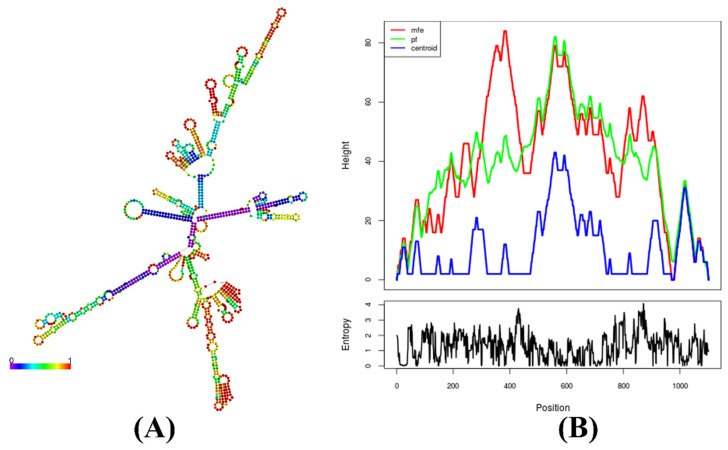
Predicted secondary structure of vaccine construct mRNA, featuring the (**A**) minimum free energy (MFE) structure and (**B**) mountain plot. This presentation highlights the relative correlation between MFE, the thermodynamic ensemble, and the centroid structure of the constructed vaccine mRNA.

**Table 1 bioengineering-11-00505-t001:** Attempted steps of subtractive proteomic analysis in *B. henselae* strain Houston-1.

S. No.	Subtractive Approaches	*B. henselae* Strain Houston-1
1	Complete set of proteins	1481
2	Mini proteins	268
3	Paralogous proteins in CD-HIT	1213
4	Non-homologs	827
5	Vital proteins in DEG	153
6	Unique metabolic pathways at KEGG	24
7	Number of vital proteins involved in KEGG and KAAS	20
8	Druggable proteins	9
9	Gut flora proteins	6
10	Cytoplasmic proteins	5
11	Membrane protein	1

**Table 2 bioengineering-11-00505-t002:** List of target proteins, including non-host homologs, subcellular localization, virulency, druggability, allergenicity, and antigenicity features.

**Protein ID**	Protein Name	Drugbank ID	Chemical Formula	Drug Name	Drug Group	Drugbank Organism	Localization	Virulency	Antigenicity Score	Antigenicity	Allergenicity
WP_011180971.1	UDP-N-acetylmuramate--L-alanine ligase	DB01673 DB03909 DB04395	C_23_H_36_N_4_O_20_P_2_ C_11_H_18_N_5_O_12_P_3_ C_10_H_17_N_6_O_12_P_3_	Uridine-5’-Diphosphate-N-Acetylmuramoyl-L-Alanine Adenosine-5’- [beta, Gamma-methylene]triphosphate Phosphoaminophosphonic Acid-Adenylate Ester	Experimental Experimental Experimental	*Haemophilus influenzae* (strain ATCC 51,907/DSM 11,121/KW20/Rd)	Cytoplasmic	Non-Virulent	0.3964	Non-Antigenic	Non-Allergic
WP_011180187.1	3-deoxy-manno-octulosonate cytidylyltransferase	DB04482	C_17_H_26_N_3_O_15_P	Cmp-2-Keto-3-Deoxy-Octulosonic Acid	Experimental	*Haemophilus influenzae* (strain ATCC 51,907/DSM 11,121/KW20/Rd)	Cytoplasmic	Virulent	0.3278	Non-Antigenic	Non-Allergic
WP_011180414.1	PAS domain-containing sensor histidine kinase (ATP-binding protein)	DB02071 DB03366	C_4_H_7_N_2_ C_3_H_4_N_2_	1-Methylimidazole Imidazole	Experimental Experimental Investigational	*Bradyrhizobium diazofficiens* strain (JCM 10833/AM 13628)	Membrane-Bound	Virulent	0.4060	Antigenic	Non-Allergic
WP_011180514.1	sigma-54-dependent Fis family transcriptional regulator	DB01857	C_4_H_8_NO_7_P	Phosphoaspartate	Experimental	*Salmonella Typhimurium* strain (CT2 1412/ATCC 700720)	Cytoplasmic	Virulent	0.3676	Non-Antigenic	Non-Allergic
WP_011180500.1	3-deoxy-8-phosphooctulonate synthase	DB01819 DB02433 DB03113 DB03936	C_3_H_5_O_6_P C_9_H_23_NO_13_P_2_ C_3_H_6_FO_6_P C_5_H_11_O_7_P	Phosphoenolpyruvate {[(2,2-Dihydroxy-Ethyl) -(2,3,4,5-Tetrahydroxy-6-Phosphonooxy-Hexyl)-Amino]-Methyl}-Phosphonic Acid 3-Fluoro-2-(Phosphonooxy)Propanoic Acid 1-Deoxy-Ribofuranose-5’-Phosphate	Experimental Experimental Experimental Experimental	*Shigella flexneri*	Cytoplasmic	Virulent	0.3274	Non-Antigenic	Non-Allergic
WP_034454605.1	Phosphoenolpyruvate--protein phosphotransferase	DB08357	C_8_H_18_O_3_	Diethylene glycol diethyl ether	Experimental	*Acinetobacter baylyi* strain ATCC 33305/ADP1)	Cytoplasmic	Non-Virulent	0.3740	Non-Antigenic	Non-Allergic

**Table 3 bioengineering-11-00505-t003:** Predicted MHC class I epitopes antigenicity, allergenicity, toxicity, and class I immunogenicity for the design of a multi-epitope vaccine construct.

T-Cell Epitopes	Antigenicity Score	Allergenicity	Toxicity	SVM	Class I Immunogenicity
AAIRFVSIY	0.8849 (Antigenic)	Non-Allergic	Non-toxic	−1.36	0.18628
ILALLYAYY	1.2102 (Antigenic)	Non-Allergic	Non-toxic	−0.77	0.01812
VTDEEELHL	1.0484 (Antigenic)	Non-Allergic	Non-toxic	−0.65	0.30924
TADGCWLKI	0.7581 (Antigenic)	Non-Allergic	Non-toxic	−0.21	0.06195

**Table 4 bioengineering-11-00505-t004:** Predicted MHC class II epitopes. The table shows the asterisk (*) in HLA-DRB signifies a specific allele variant within the HLA-DRB gene, commonly used in HLA nomenclature to denote subtype. Furthermore, the epitopes were checked for antigenicity, allergenicity, and toxicity for the design of a multi-epitope vaccine construct.

MHC-II Peptide	Start	HLA Alleles	Antigenicity Score	Allergenicity	Toxicity
ALLYAYYKTDSISEK	39	HLA-DRB1 * 04:05	0.4707 (Antigenic)	Non-Allergic	Non-Toxic
IALSHTYISEKTQEI	21	HLA-DRB3 * 01:01	0.4331 (Antigenic)	Non-Allergic	Non-Toxic
KTDSISEKIRAMYEM	46	HLA-DRB1 * 13:02	0.6838 (Antigenic)	Non-Allergic	Non-Toxic
LLYAYYKTDSISEKI	40	HLA-DRB1 * 04:05	0.5207 (Antigenic)	Non-Allergic	Non-Toxic
YKTDSISEKIRAMYE	45	HLA-DRB1 * 13:02	0.4995 (Antigenic)	Non-Allergic	Non-Toxic

**Table 5 bioengineering-11-00505-t005:** Predicted linear B-cell epitopes, their antigenicity, allergenicity, and toxicity analysis.

B-Cell Peptide	Antigenicity Score	Antigenicity	Allergenicity	Toxicity
KSSAQNHKARTKHINP	0.6231	Antigenic	Non-Allergic	Non-toxic
TSIRQTADGCWLKINE	0.6391	Antigenic	Non-Allergic	Non-toxic
LAAIRFVSIYDLRHTI	0.6559	Antigenic	Non-Allergic	Non-toxic
KLEIISKEMKGTTVTI	0.6320	Antigenic	Non-Allergic	Non-toxic
KEMKGTTVTITMPIKQ	0.4484	Antigenic	Non-Allergic	Non-toxic
NQLTKTHTGSGLGLAI	1.0892	Antigenic	Non-Allergic	Non-toxic
RHTIDKNTRSTITLLA	2.2375	Antigenic	Non-Allergic	Non-toxic

**Table 6 bioengineering-11-00505-t006:** Physiochemical properties of the designed multi-epitope vaccine construct.

Physiochemical Features	Evaluation
Amino acid residue	367
Molecular weight	38.84 kDa
Theoretical PI	9.34
Total number of negatively charged residue (Asp + Glu)	28
Total number of positively charged residues (Arg + Lys)	44
Formula	C_1737_H_2748_N_472_O_509_S_13_
Extinction coefficients	97,000 M^−1^ Cm^−1^
Estimated half-life	30 h (mammalian reticulocytes, in vitro) >20 h (Yeast, in vitro) >10 h (*Escherichiaa coli*, in vivo)
Instability index	32.68 (stable)
Aliphatic index	75.59
Gravy	−0.377

## Data Availability

The datasets to support the conclusions of this article are given within the article.

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
