# Peer review of "Subtractive Proteomics and Reverse-Vaccinology Approaches for Novel Drug Target Identification and Chimeric Vaccine Development against Bartonella henselae Strain Houston-1"

_bioengineering, 2024, doi:10.3390/bioengineering11050505_

Round 1
Reviewer 1 Report
Comments and Suggestions for Authors
The manuscript is devoted to the search for drug targets and the design of an artificial polypeptide immunogen for a vaccine against Bartonella henselae according to the principles of the subtractive proteomics and the reverse vaccinology. This work represents a profound in silico design that comprises almost all aspects of the immunogen construction, preparation and immunogenicity prediction. The drug target search is described in less details. However, despite of the profound character of the manuscript, here are some notes concerning the presentation of the research and some questions that arise to its results and conclusions.
1. The manuscript title is somewhat confusing. Firstly, the constructed and described multi-epitope artificial protein is not a vaccine, but an immunogen, or maybe, a vaccine construct. It will obviously require a certain support substance to ensure its stability and gradual release from an injection site in order to attract immunocompetent cells for a period of time. Secondly, the results of the drug target search are not described in details: what about the perspectives of the search for possible drug substances (inhibitors, other regulating compounds) for these targets? Besides that, the results of the drug target search are not properly reflected in the Conclusion section. What about the concentration on the description of the immunogen design?
2. Since the authors employ the reverse vaccinology approach to the immunogen design, at least one citation of the paper describing its principles is needed.
3. Materials and Methods are not properly described. The Results section contains a lot of citations that concern the detalized description of the used methods. These citations and corresponding details should be moved to the Materials and Methods section.
Some details: a) Line 102: CD-HIT is a server; an exact indication fo the program used is required.
b) Line 200: a source for the GalaxyRefine tool access is required; abbreviation for ERRAT should be deciphered.
c) Lines 223-223: abbreviations are not deciphered (except GROMACS).
4. Supplementary figure 3 graphs are too small and hence not well understandable; they should be made larger.
Supplementary table 1: what does two-component system mean?
Supplementary table 3: the presentation of the peptide fragments is almost unreadable. It should be of the kind: 10AAAA...AA80.
5. The authors' approach to the design of the vaccine construct poses a question to the location of B- and T-helper epitopes. It is known that an adjacent location of B- and T-helper epitopes (and even their overlap) is required for a more efficient antibody production. However, the authors have put the predicted B- and T-helper epitopes in different parts of the artificial construct. It should be well explained, why.
6. Regarding the discontinuous B-epitopes. It is incorrect to prescribe their location to the whole long peptide fragments. Molecular modeling could define at least some amino acid residues in these fragments that are located enough close to one another to form a putative discontinuous B-epitope that can correspond to an antibody paratope size.
7. The data in Figure 2 and Table 7 do not correlate in terms of helix contents - it should be explained. Regarding the Figure 2 - more description is required in the legend. Besides that, it is better to show the location of epitopes in the construct structure in this figure.
Regarding Cys residues in the construct: do they form S-S- bridges or not?
8. Line 454-455: What does it mean:"nonbonding interactions"? Could these be defined more precisely?
9. Does the term "stability" mean the stability of the spatial structure in a solution? What about the stability of the vaccine construct to protease digestions? Can the predicted cytotoxic T-epitopes be properly formed by proteasome digestion?
10. Regarding the simulation of immunity. The initial points for this simulation (immunogen dosage, number of injections, construct stability to proteases, its deposition in the injection site) are not defined; hence it does not allow evaluating whether this simulation is correct.
Comments on the Quality of English LanguagePlease correct some typing errors.
Author Response
- The manuscript title is somewhat confusing. Firstly, the constructed and described multi-epitope artificial protein is not a vaccine, but an immunogen, or maybe, a vaccine construct. It will obviously require a certain support substance to ensure its stability and gradual release from an injection site in order to attract immunocompetent cells for a period of time. Secondly, the results of the drug target search are not described in details: what about the perspectives of the search for possible drug substances (inhibitors, other regulating compounds) for these targets? Besides that, the results of the drug target search are not properly reflected in the Conclusion section. What about the concentration on the description of the immunogen design?
Reply: Dear reviewer, thank you for your comments. We have revised the manuscript title and that the multi-epitope artificial protein described in our manuscript serves as an immunogen or a vaccine construct. To address the concerns regarding stability and controlled release, we have incorporated Beta defensin as an adjuvant in our vaccine formulation which is mentioned in lines 199, 201-203 and in lines 451-453 in the revised manuscript. Beta defensin is recognized for its capacity to stabilize antigens and facilitate controlled release from the injection site, ensuring sustained exposure to immunocompetent cells. Regarding the search for drug target substance, our primary focus was on identifying novel drug targets for the Bartonella pathogen and designing a multi-epitope vaccine. The objective of our paper was not to investigate drug inhibitors or regulating compounds, but rather to lay the foundation for future research in this area. Besides the results of the drug target, we expanded the conclusion to include the drug target results in lines 629-632. The manuscript thoroughly describes the immunogen design process. We discuss the selection criteria for epitopes, the rationale behind adjuvants and linkers, and the physicochemical properties of the vaccine construct. In-depth analyses like molecular docking and structural simulations are included, providing a comprehensive account of our approach.
- Since the authors employ the reverse vaccinology approach to the immunogen design, at least one citation of the paper describing its principles is needed.
Reply: Thank you for your comment. We have provided a citation in the manuscript regarding the principles of reverse vaccinology for immunogen design in lines 91 and 93, 94 of the revised manuscript.
- Materials and Methods are not properly described. The Results section contains a lot of citations that concern the detalized description of the used methods. These citations and corresponding details should be moved to the Materials and Methods section.
Reply: Thank you for your comment. We made changes to methods and materials to describe it properly, and also moved some citations and corresponding details from the Results section to the Materials and Methods section in the revised manuscript.
Some details: a) Line 102: CD-HIT is a server; an exact indication for the program used is required.
- b) Line 200: a source for the GalaxyRefine tool access is required; abbreviation for ERRAT should be deciphered.
- c) Lines 223-223: abbreviations are not deciphered (except GROMACS).
Reply: Thank you for your valuable suggestions and for spotting these mistakes (a) We have included the exact indication for the CD-HIT server which is mentioned in line 114, 115 in the revised manuscript. (b) We have provided the source for accessing the Galaxy Refine tool in line 219, 220, as well as an abbreviation for ERRAT in the revised manuscript which is mentioned in line 220, 221. (c) we have deciphered all the abbreviations. The changes have been made in the revised manuscript specifically mentioned in lines 243, 254, 255-256, 259-260, 261.
- Supplementary Figure 3 graphs are too small and hence not well understandable; they should be made larger.
Supplementary table 1: what does a two-component system mean?
Supplementary table 3: the presentation of the peptide fragments is almost unreadable. It should be of the kind: 10AAAA...AA80.
Reply: Thank you, we have increased the size of Supplementary Figure 3 graphs. In supplementary Table 1, the term 'two-component system' refers to a signaling mechanism commonly found in bacteria. It involves two key components - a sensor kinase and a response regulator. The sensor kinase detects environmental signals and phosphorylates itself, then transfers the phosphate group to the response regulator, triggering a cellular response. We have updated Supplementary Table 3, which is mentioned in Supplementary Table S7, as well as in the manuscript legend (Figure 6) on lines 571-5745.
- The authors' approach to the design of the vaccine construct poses a question to the location of B- and T-helper epitopes. It is known that an adjacent location of B- and T-helper epitopes (and even their overlap) is required for a more efficient antibody production. However, the authors have put the predicted B- and T-helper epitopes in different parts of the artificial construct. It should be well explained, why.
Reply: Dear reviewer, thank you for your comment regarding the positioning of B- and T-helper epitopes in our vaccine construct. While it is established that adjacent positioning of these epitopes can enhance antibody production, we strategically placed them in different parts of the construct. This decision aimed to promote a broader immune response, targeting multiple epitopes to increase the likelihood of capturing a diverse range of antigen-specific T and B cells. Additionally, this approach minimizes the risk of immune dominance and potential self-reactivity.
- Regarding the discontinuous B-epitopes. It is incorrect to prescribe their location to the whole long peptide fragments. Molecular modeling could define at least some amino acid residues in these fragments that are located enough close to one another to form a putative discontinuous B-epitope that can correspond to an antibody paratope size.
Reply: Thank you for your comment. We have revised the figure of discontinuous B-cell epitopes to clarify the location of epitopes, showing amino acid residues in these fragments that are close enough to form a putative discontinuous B-epitope, as mentioned in Figure 6.
- The data in Figure 2 and Table 7 do not correlate in terms of helix contents - it should be explained. Regarding the Figure 2 - more description is required in the legend. Besides that, it is better to show the location of epitopes in the construct structure in this figure.
Regarding Cys residues in the construct: do they form S-S - bridges or not?
Reply: Dear reviewer, thank you for your comment. We generated the secondary structure using PDBsum, while the alpha, beta, and coil structures were predicted by SOPMA. However, considering the importance of accuracy, we replaced the SOPMA results with those from PDBsum. PDBsum generates secondary structure using protein structures from the Protein (PDB file), ensuring higher accuracy. The updated results are included in Supplementary Table 6. This ensures that our analysis aligns with experimental data, enhancing the credibility of our findings. Regarding the figure 2 legend, we provide more description which is mentioned in lines 478-481.Regarding Cys residues in the construct, a total of 32 residue pairs were assessed, with only 2 pairs meeting the specific criteria for modification. These pairs underwent mutations, resulting in the formation of four disulfide bonds. We performed Disulfide engineering to stabilize the structural integrity of the construct which is mentioned in the revised manuscript in lines 226-232 and lines 511-520 and supplementary figure S4.
- Line 454-455: What does it mean:" nonbonding interactions"? Could these be defined more precisely?
Reply: Dear reviewer, Nonbonding interactions, also known as van der Waals interactions, refer to the weak forces of attraction between atoms or molecules that are not involved in covalent bonding. These interactions include dispersion forces, dipole-dipole interactions, and hydrogen bonding. In the context of our study, nonbonding interactions play a crucial role in stabilizing the three-dimensional structure of proteins and facilitating molecular recognition events which are predicted by the PDBsum tool.
- Does the term "stability" mean the stability of the spatial structure in a solution? What about the stability of the vaccine construct to protease digestions? Can the predicted cytotoxic T-epitopes be properly formed by proteasome digestion?
Reply: The term "stability" used in our context encompasses both the structural integrity of the vaccine construct in a solution and its resistance to protease digestions. Specifically, it refers to the vaccine construct's ability to maintain its folded conformation in solution and its resistance to degradation by proteases. Furthermore, we have predicted CTL epitopes by combining MHC-binding peptide prediction, proteasome cleavage site prediction, and TAP transport efficiency prediction to ensure the proper formation of cytotoxic T-epitopes during proteasome digestion which is mentioned in the supplementary table S3.
- Regarding the stimulation of immunity. The initial points for this simulation (immunogen dosage, number of injections, construct stability to proteases, its deposition in the injection site) are not defined; hence it does not allow evaluating whether this simulation is correct.
Reply: Dear reviewer, we have defined the initial parameters for the simulation of immunity, including immunogen dosage, number of injections, construct stability to proteases, and its deposition in the injection site. These details are provided in the revised manuscript in lines 269-276 and lines 577-582.
Reviewer 2 Report
Comments and Suggestions for Authors
The authors used reverse-vaccinology-based proteomics to explore and design a multi-epitope vaccine against the B. henselae strain Houston-1. The PAS domain-containing sensor histidine kinase protein was selected, and several epitopes based on this protein were evaluated. The antigenicity, allergenicity, solubility, MHC binding capability, and toxicity of these vaccines were assessed and checked for their effectiveness against B. henselae. Importantly, this manuscript describes a new approach and convincingly demonstrates its effectiveness, paving the way for future research in this area.
1. The background does not provide efficient information, particularly the substractive proteomic and reverse vaccinology.
2. The manuscript contains many typos. For example, the figure lengend in figure 5. “ Molecular dynamics simulation for vaccine-TLR complex (A). Root Mean Square 477 Deviation (RMSD), (B). Root Mean Square Fluctuation (RMSF), (C). Radius of gyration 478 (Rg), (D). Represents 0 ns and 100 ns frames from Vaccine construct simulation, (E). 479 Represents 0 ns and 100 ns frames from TLR4 simulation, (F), (E). Represents 0 ns and 100 480 ns frames from Vaccine+TLR4 simulation.”
3. Some important parameters, such as molecular docking (cluspro) and 3D structure prediction (3Dpro), were not described in context.
4. The score of cluspro docking should be reported.
5. The authors used one method to create 3D structure. Using another method, such as Alpgafold2, can further verify the structure model
Comments on the Quality of English LanguageThe manuscript contains many typos.
Author Response
- The background does not provide efficient information, particularly the substractive proteomic and reverse vaccinology.
Reply: Dear reviewer, thank you for your comment. We've improved the background by adding more details about subtractive proteomics and reverse vaccinology methods, as mentioned on lines 81-96 in the revised manuscript.
- The manuscript contains many typos. For example, the figure lengend in figure 5. “Molecular dynamics simulation for vaccine-TLR complex (A). Root Mean Square 477 Deviation (RMSD), (B). Root Mean Square Fluctuation (RMSF), (C). Radius of gyration 478 (Rg), (D). Represents 0 ns and 100 ns frames from Vaccine construct simulation, (E). 479 Represents 0 ns and 100 ns frames from TLR4 simulation, (F), (E). Represents 0 ns and 100 480 ns frames from Vaccine+TLR4 simulation.”
Reply: Dear reviewer, thank you for spotting these typos. We have corrected typos in the manuscript, specifically in the figure legend for Figure 5, and revised the legend accordingly, which is mentioned in lines 558-565.
- Some important parameters, such as molecular docking (cluspro) and 3D structure prediction (3Dpro), were not described in context.
Reply: We have expanded our manuscript to include additional information on the key parameters. Specifically, details on ClusPro can be found in lines 236-239 and lines 527-529. Likewise, information on 3Dpro is available on lines 215-218 and lines 484-489.
- The score of cluspro docking should be reported.
Reply: Dear reviewer, the scores of the top 10 ClusPro dockings are reported in the revised manuscript on lines 531-533.
- The authors used one method to create 3D structure. Using another method, such as Alpfafold2, can further verify the structure model.
Reply: Dear reviewer, thank you for your suggestion. Firstly, we tried to model with AlphaFold, and the predicted structure was not reliable as all the top 5 structures predicted were having below 40 predicted IDDT (pIDDT). From the literature, it has been known that the predicted structure below 70 pIDDT cannot be reliable. Because of this, we opted to predict the structure with 3Dpro, considering its compatibility with our construct, and stand by its accuracy as well as reported for many vaccine constructs from the available literature.
Reviewer 3 Report
Comments and Suggestions for Authors
Bartonella henselae is the causative agent of cat-scratch disease and is primarily transmitted by infected fleas. Despite progress in understanding its pathogenesis, no definite treatment regimen is known for a patient infected with B. henselae due to limited knowledge about the virulence factors and regulatory mechanisms specific to the B. henselae strain. To fill this gap, the authors used reverse-vaccinology-based subtractive proteomics to identify novel targets for multi-epitope vaccine development to combat B. henselae infections. They identified a membrane protein, the PAS domain-containing sensor histidine kinase protein, as an antigenic protein and created a multi-epitope vaccine construct in which filtered B cell and T cell epitopes for the PAS domain-containing sensor histidine kinase protein were merged using linkers and an adjuvant. They demonstrated the favorable structural, physicochemical, and immunological attributes of the proposed vaccine and claimed that the immune simulations indicated promising in-vivo immunogenicity.
Here are some minor issues that should be addressed before acceptance.
1. For Figure 1, it would be better to label the protein names or numbers in each section in Figure B. It is better to exchange the order of Figures 1A and 1B based on the order they appear in the text.
2. For all figures, it would be more readable to label A,B,C... at the upper left corner of each figure sub-section.
3. The full list of 1) 14 epitopes within the PAS domain-containing sensor histidine kinase, 2) 55 MHC-II epitopes, and 3) 19 linear B-cell epitopes should be provided in the supplementary information.
4. In Figure 3, the figure legend for the vaccine sequence was missing. And the information represented by different colors was incomplete.
5. In Figure 4, for a better illustration, the name of TLR4 and vaccine should be labeled in 4A and 4B.
Author Response
- For Figure 1, it would be better to label the protein names or numbers in each section in Figure B. It is better to exchange the order of Figures 1A and 1B based on the order they appear in the text.
Reply: Thanks for your suggestion. We labeled the protein number in each section of Figure 1A, and also exchanged its legends and order based on what appeared in the text.
- For all figures, it would be more readable to label A, B, C... at the upper left corner of each figure sub-section.
Reply: Thanks for your suggestion. We have added labels at the upper left corner of each figure sub-section for better readability.
- The full list of 1) 14 epitopes within the PAS domain-containing sensor histidine kinase, 2) 55 MHC-II epitopes, and 3) 19 linear B-cell epitopes should be provided in the supplementary information.
Reply: The full list of MHC-I, MHC-II, and linear B-cell epitopes was provided in the supplementary Tables S3, S4, and S5.
- In Figure 3, the figure legend for the vaccine sequence was missing. And the information represented by different colors was incomplete.
Reply: Thank you. We have revised Figure 3 to include the legend for the vaccine sequence and corrected the information represented by different colors, as mentioned in lines 502-504.
- In Figure 4, for a better illustration, the name of TLR4 and vaccine should be labeled in 4A and 4B.
Reply: Thank you for the suggestion. We have added labels to "TLR4" and "Vaccine construct" in Figures 4A and 4B in the revised manuscript.
Round 2
Reviewer 1 Report
Comments and Suggestions for Authors
All previous comments have been replied, corrections have been made. No additional comments.
Comments on the Quality of English LanguageMinor corrections in article use may be required.
Reviewer 2 Report
Comments and Suggestions for Authors
The revised manuscript has answered most of the questions.